# Plasticity by Precision: Exemplar-free Analytic Adaptation for Class-Incremental Learning

## Abstract

Class-Incremental Learning (CIL) aims to enable models to acquire new knowledge sequentially while preserving previously learned information, emulating human-like learning capabilities. Current methods, including pre-trained foundation models and Experience Replay (ER) methods, serve as strong baselines for sequential task learning. However, these methods remain prone to catastrophic forgetting, especially in online settings with non-stationary data and blurry task boundaries. Additionally, the requirement to store historical samples in ER-based methods introduces significant memory overhead and privacy risks, limiting the practical adoption of CIL models in real-world applications. To address this, we propose an **E**xemplar-**f**ree **A**nalytic **A**daptation for **C**lass-**I**ncremental **L**earning (AACL) framework that updates the classifier in a principled probabilistic manner. Our key contribution is a closed-form Bayesian update that unifies three critical components: (1) the prior precision encapsulating knowledge from previous tasks, (2) a Fisher Information-inspired weight penalty to protect learned knowledge, and (3) the feature correlation matrix representing evidence from new data. Our framework balances plasticity and stability by integrating prior knowledge with streaming data, preserving learned representations while adapting to new tasks. We conduct comprehensive evaluations on benchmark datasets under the SI-Blurry setting, achieving $\mathcal{A}_{\mathrm{AUC}}$ improvements of 8%, 3%, and 4% on CIFAR-100, ImageNet-R, and Tiny-ImageNet, respectively.

## 1 Introduction

In recent years, Deep Learning (DL) methods have achieved remarkable success across domains such as vision, speech, and language through large-scale supervised training on static datasets Cao et al. (2024a); Zhu et al. (2025). However, this conventional paradigm falls short in dynamic, real-world environments where data distributions shift over time and new class concepts emerge continuously. Class-Incremental Learning (CIL) addresses this challenge by enabling models to sequentially learn novel classes without access to data from prior tasks Rebuffi et al. (2017); Zhou et al. (2024). A key challenge in this setting is catastrophic forgetting, where learning new information leads to the degradation of previously acquired knowledge. This forgetting is further exacerbated in exemplar-free settings, where models cannot store or revisit prior data due to privacy or memory limitations. To mitigate the forgetting problem, recent CIL research has increasingly focused on leveraging Pre-Trained Models (PTMs) Zhuang et al. (2024a); Gao et al. (2025b), which provide strong generalization capabilities due to their training on large and diverse datasets. Instead of retraining entire networks from scratch Li & Hoiem (2017), modern approaches freeze PTMs and attach lightweight modules that adapt to incoming tasks Moon et al. (2023); Sun et al. (2025). Among these, prompt-based methods have gained traction due to their efficiency and adaptability. For instance, L2P Wang et al. (2022b) and DualPrompt Wang et al. (2022a) dynamically select task-relevant prompts from a pool to guide feature adaptation, while CODA-Prompt Smith et al. (2023) improves prompt retrieval via attention-based selection. Despite these advancements, these methods still suffer from forgetting at two critical levels: parameter drift that degrades learned representations and retrieval errors from suboptimal prompt selection during inference.

Recent work has explored various strategies to address these challenges. Some methods aim to stabilize learning by orthogonalizing prompts or enforcing inter-task separation Hu et al. (2024); Liang & Li (2024),

while others turn to replay buffers that store previous samples Moon et al. (2023). However, these approaches often introduce considerable memory overhead or remain susceptible to task-recency bias. This problem is particularly pronounced in generalized class-incremental learning (GCIL) settings where class distributions are imbalanced across tasks Zhuang et al. (2024b). Analytical Continual Learning (ACL) methods provide a promising alternative to gradient-based approaches by enabling closed-form updates that precisely preserve prior knowledge Zhuang et al. (2024a); Qi et al. (2024). However, existing ACL models typically assume well-separated tasks and fail to account for class overlap, retrieval noise, or distributional shifts commonly encountered in blurry-boundary settings. Furthermore, real-world continual learning settings often involve ambiguous task boundaries, overlapping class distributions, and strict data privacy constraints, where traditional methods struggle to scale or generalize effectively. These challenges motivate the need for a robust, exemplar-free framework that balances the stability-plasticity trade-off while remaining computationally efficient and privacy-preserving.

The scope of this research focuses on class-incremental continual learning, where AACL integrates Fisher-based regularization with Bayesian linear regression to derive a closed-form posterior update rule. Unlike conventional Bayesian models that rely on gradient-based inference or variational approximations, AACL employs analytic precision-weighted updates that follow the weight-invariant property of ACL (Zhuang et al. (2024a); Momeni et al. (2025)), where the recursively accumulated precision matrix and prior capture statistics bound to the backbone's embedding distribution. This ensures that knowledge accumulated across tasks is preserved exactly without replay or gradient regularization, resulting in a robust and scalable solution for class-incremental continual learning. Our key contributions are as follows:

- We propose AACL, an exemplar-free analytic adaptation framework for continual learning that learns effectively in task-agnostic environments with ambiguous task boundaries, achieving robust sequential learning without relying on historical data or replay mechanisms.

- We introduce a novel unified precision matrix that integrates prior precision, Fisher-informed knowledge retention, and current-task covariance structure to provide an efficient and scalable solution for preventing catastrophic forgetting in dynamic environments (e.g., blurry tasks)

- AACL formulates a closed-form Bayesian update that maintains a full-covariance Gaussian posterior over classifier weights, enabling continual adaptation without exemplars or task boundaries. Our extensive evaluations on CIFAR-100, ImageNet-R, and Tiny-ImageNet under the Si-Blurry setting show $\mathcal{A}_{\mathrm{AUC}}$ gains of 8%, 3%, and 4%, respectively.

## 2 Related Work

Class-Incremental Learning (CIL) aims to enable models to continually learn from sequentially arriving tasks without accessing historical data, while maintaining performance on previously seen classes. The central challenge in CIL is catastrophic forgetting, where neural networks lose previously acquired knowledge when adapting to new tasks. To address this limitation, researchers have developed a wide range of methodological approaches. Regularization-based methods address forgetting by penalizing changes to important weights, as measured by task-specific importance metrics such as the Fisher Information Zenke et al. (2017); Chen & Zhou (2025). These approaches treat learning as a stability-plasticity trade-off, where parameters crucial for prior tasks are regularized to remain stable. Replay-based methods, on the other hand, retain a small buffer of past examples Liu et al. (2023); Aghasanli et al. (2025) or use generative models to simulate historical data Bellitto et al. (2024); Ye & Bors (2025), allowing the model to interleave new data with representative samples of earlier tasks. Distillation-based methods preserve knowledge by aligning the outputs or intermediate representations of the current model with those of a previous version Szatkowski et al. (2024); Gao et al. (2025a). More recently, the rise of large-scale PTMs has opened new opportunities for CL. Parameter-Efficient Fine-Tuning techniques Cao et al. (2024b); Sun et al. (2025); KANG et al. (2025) modify only a small subset of parameters, such as adapter layers or low-rank matrices, while keeping the core model frozen. These methods are particularly effective when storage and computational budgets are constrained, as they avoid catastrophic forgetting by isolating task updates.

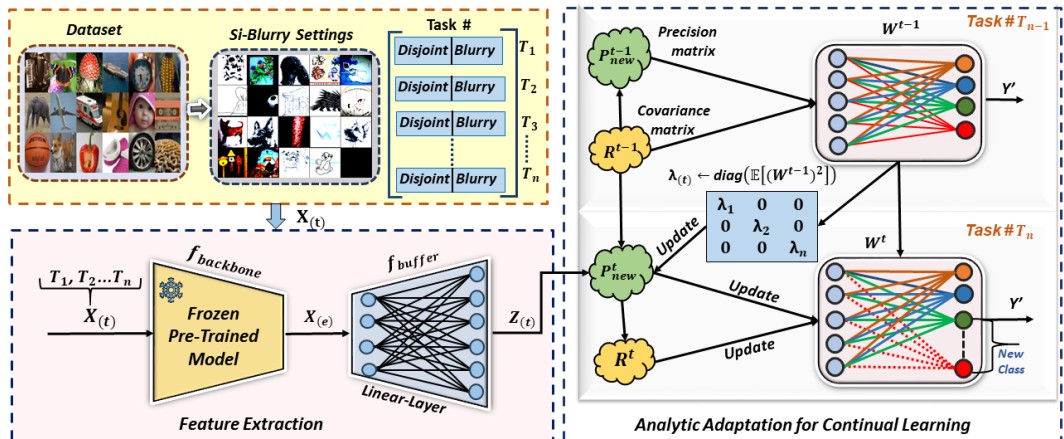

Figure 1: **An overview of the AACL Framework for Analytic Continual Learning.** AACL utilizes a *frozen pre-trained backbone* $f_{\text{backbone}}$ to extract feature representations $X_{(e)}$, which are subsequently transformed through a *randomly initialized, non-trainable buffer layer* $f_{\text{buffer}}$ to generate adapted features $Z_{(t)}$. AACL analytically computes task-specific classifier weights $W^t$ using closed-form solutions based on the *feature covariance matrix* $R^{(t)}$ and *precision matrix* $P_{new}^{(t)}$, thereby achieving scalable, privacy-preserving, and computationally efficient continual learning without buffer training or exemplar storage.

**Analytic Continual Learning (ACL).** has emerged as a principled, efficient alternative to gradient-based approaches. ACL methods bypass stochastic optimization entirely by framing learning as a closed-form solution to a regularized least-squares problem Zhuang et al. (2022); McDonnell et al. (2023). This formulation enables fast, stable updates through recursive matrix computations that accumulate task-level statistics. Subsequent extensions have incorporated geometric constraints (e.g., residual projection Qi et al. (2024)) and subspace alignment Zhou et al. (2024) to enhance generalization across tasks. GACL Zhuang et al. (2024a) proposes a gradient-free, closed-form solution for Generalized Class Incremental Learning (GCIL) without using exemplars, addressing privacy concerns. It leverages a weight-invariant property to achieve equivalence with joint training, improving performance across tasks. AnaCP Momeni et al. (2025) achieves feature adaptation through a learnable projection layer trained using contrastive learning. MoAL Gao et al. (2025b) proposes a momentum-based analytical learning framework that enhances class-incremental learning by combining a closed-form classifier with adaptive adapter interpolation. It further introduces a knowledge rumination mechanism to revisit and reinforce old knowledge, balancing plasticity and stability. Despite their benefits, many ACL algorithms assume disjoint task labels, which restricts their applicability in realistic CIL scenarios involving shared classes, partial overlaps, or blurred task boundaries. To address these challenges, we introduce a robust exemplar-free framework that handles overlapping class distributions and adapts seamlessly to blurry task boundaries.

# 3 Proposed Method

This section presents the AACL framework for class-incremental continual learning. We begin with the problem formulation, followed by a detailed description of the proposed analytic update mechanism and its core design principles. Additional details on improving computational efficiency via Woodbury updates are provided in the Appendix.

**Problem Definition.** We denote the entire dataset by $\mathcal{S}$. When $\mathcal{S}$ is divided into a sequence of tasks, we define the training dataset at task $t$ as $\mathcal{S}_t^{\text{train}} = \{\mathbf{X}_t^{\text{train}}, \mathbf{Y}_t^{\text{train}}\}$, where $\mathbf{X}_t^{\text{train}} \in \mathbb{R}^{N_t \times C \times W \times H}$ consists of $N_t$ input images with channel size $C$ and spatial dimensions $W \times H$. The corresponding labels are represented as $\mathbf{Y}_t^{\text{train}} \in \mathbb{R}^{N_t \times d_t}$, where $d_t$ is the number of unique classes observed up to task $t$. The test set for task $t$ is denoted by $\mathcal{S}_t^{\text{test}} = \{\mathbf{X}_t^{\text{test}}, \mathbf{Y}_t^{\text{test}}\}$. The goal of continual incremental learning at task $t$ is to train models

using $\mathcal{S}_t^{\text{train}}$ and evaluate their performance on the accumulated test data $\mathcal{S}_{1:t}^{\text{test}} = \cup_{i=1}^{t} \mathcal{S}_i^{\text{test}}$. Following the GCIL framework Mi et al. (2020), we reformulate the Si-Blurry scenario for AACL.

**Property 1.** *The number of classes in a task is not fixed.* Suppose $d_t$ is the number of classes in task $t$, we have:

$$d_t = \left|\left\{j \in \mathcal{Y} : y_{t,j}^{\text{train}} > 0\right\}\right| \sim \mathcal{D}_t \tag{1}$$

where $\mathcal{D}_t$ is a task-dependent distribution.

**Property 2.** *Classes appearing in different tasks could overlap.* For two tasks $t$ and $t'$, $t \neq t'$, we have:

$$\mathbb{P}\left(Y_t^{\text{train}} \circ Y_{t'}^{\text{train}} \neq 0\right) > 0 \tag{2}$$

where $\circ$ denotes element-wise multiplication of two label vectors and $\mathbb{P}(\cdot)$ is the probability.

**Property 3**. *Sample sizes of different classes at the same task could be different.* That is, for task $t$, we have:

$$i, j \in \mathcal{Y}, \ i \neq j, \ \mathbb{P}\left(y_{t,i}^{\text{train}} \neq y_{t,j}^{\text{train}} \mid y_{t,i}^{\text{train}} > 0, \ y_{t,j}^{\text{train}} > 0\right) > 0 \tag{3}$$

To effectively utilize pre-trained knowledge, we adopt a frozen feature extractor $\psi_{\text{backbone}}$, such as a Vision Transformer (ViT) Touvron et al. (2021), to encode input images into an intermediate representation. Given an input $\mathbf{X}_{(\mathbf{t})}$, the extracted embedding is defined as:

$$\mathbf{X}_{(e)} = f_{\text{b}}(\mathbf{X}_{(\mathbf{t})}; \psi_{\text{backbone}}), \tag{4}$$

where $f_{\text{b}}(\cdot)$ denotes the frozen backbone feature extractor. To further adapt the extracted embeddings for downstream continual tasks, we introduce a buffer transformation layer $f_{\text{buffer}}(\cdot)$ that maps the backbone embeddings into a buffered representation: $\mathbf{Z}_{(t)} = f_{\text{buffer}}(\mathbf{X}_{(e)})$. Several options exist for $f_{\text{buffer}}$, such as a random projection layer used in ACIL Zhuang et al. (2022) or kernel-based mappings as seen in GKEAL Zhuang et al. (2023). In our setup, we follow the formulation in ACIL and define the buffer transformation as:

$$f_{\text{buffer}}(\mathbf{X}_{(e)}) = \phi(\mathbf{X}_{(e)}, \mathbf{W}_{\mathbf{buffer}}), \tag{5}$$

where $\mathbf{W}_{\text{buffer}}$ denotes the buffer layer weights randomly sampled from a normal distribution, and $\phi(\cdot)$ represents a non-linear activation function (e.g., ReLU).

### 3.1 Analytic Adaptation for Continual Learning

We propose an Analytic Adaptation for Continual Learning (AACL) framework as represented in Figure 1, inspired by the Bayesian regression model, to enable exemplar-free continual learning with closed-form updates. At task $t$, given the previously learned weights $\mathbf{W}^{(t-1)}$, precision matrix $\mathbf{P}^{(t-1)}$, and the incoming data $(\mathbf{X}_{(t)}, \mathbf{Y}_{(t)})$. We introduce a diagonal approximation of the Fisher information matrix Nguyen et al. (2018); Kirkpatrick et al. (2017) to estimate parameter importance and regularize updates. We approximate the Fisher information matrix using the squared magnitudes of previously learned weights. This approximation acts as a regularizer to retain prior knowledge while adapting to new tasks.

$$\boldsymbol{\lambda}_{\text{fisher}} = \text{diag}\left(\mathbb{E}[(\mathbf{W}^{(t-1)})^2]\right) = \text{diag}\left(\frac{1}{c}\sum_{j=1}^{c}(\mathbf{W}_{:,j}^{(t-1)})^2\right) \tag{6}$$

where $c$ is the number of output classes, and $\mathbf{W}_{:,j}^{(t-1)}$ denotes the $j$-th column of the weight matrix from the previous task.

**Key insight**. Traditional gradient-based Fisher information methods (e.g., EWC) suffer from learning instability due to ambiguous, noisy gradient directions caused by overlapping or blurry tasks. To address this, we empirically found that squared weights provide a deterministic measure of parameter importance, capturing stable precision while preserving long-term confidence to mitigate catastrophic forgetting.

---

**Algorithm 1** The pseudo-code of AACL

---

**Require:** Sequence of tasks $\mathcal{S} = \{\mathbf{X}_{(t)}, \mathbf{Y}_{(t)}\}_{t=1}^{T}$, frozen backbone $f_{\mathrm{b}}$ with weights $\boldsymbol{\psi}_{\mathrm{backbone}}$, buffer layer $f_{\mathrm{buffer}}$, hyperparameters $\gamma$, $\eta$, $\beta$, $\varepsilon$.
1: **Initialize:** $\mathbf{R}^{0} \leftarrow \gamma\mathbf{I}$, $\mathbf{W}^{0} \leftarrow \mathbf{0}$, $\quad \{\mathbf{R}^{0} \in \mathbb{R}^{D \times D}, \mathbf{W}^{0} \in \mathbb{R}^{D \times C}\}$
2: **for** $t = 1$ **to** $T$ **do**
3: $\quad \mathbf{X}_{(e)} \leftarrow f_{\mathrm{b}}(\mathbf{X}_{(t)}; \boldsymbol{\psi}_{\mathrm{backbone}}), \quad \{\mathbf{X}_{(t)} \in \mathbb{R}^{B \times D_{emb}}\}$
4: $\quad \mathbf{Z}_{(t)} \leftarrow f_{\mathrm{buffer}}(\mathbf{X}_{(e)}), \quad \{\mathbf{Z}_{(t)} \in \mathbb{R}^{B \times D}\}$
5: $\quad \boldsymbol{\lambda}_{(t)} \leftarrow \mathrm{diag}\big(\mathbb{E}[(\mathbf{W}^{t-1})^{2}]\big)$
6: $\quad \mathbf{P}^{(t)} \leftarrow \big(\mathbf{R}^{(t-1)}\big)^{-1} + \eta\,\boldsymbol{\lambda}_{(t)} + \beta\,\mathbf{Z}_{(t)}^{\top}\mathbf{Z}_{(t)}, \quad \{\textit{Precision update}, \mathbf{P}^{(t)} \in \mathbb{R}^{D \times D}\}$
7: $\quad \mathbf{R}^{(t)} \leftarrow \big(\mathbf{P}^{(t)} + \epsilon\,\mathbf{I}\big)^{-1}, \quad \{\textit{Posterior covariance}, \mathbf{R}^{(t)} \in \mathbb{R}^{D \times D}\}$
8: $\quad \mathbf{m}_{\mathrm{prior}} \leftarrow \Big(\big(\mathbf{R}^{(t-1)}\big)^{-1} + \eta\,\boldsymbol{\lambda}_{(t)}\Big)\mathbf{W}^{(t-1)}, \quad \{\textit{Prior component}\}$
9: $\quad \mathbf{m}_{\mathrm{data}} \leftarrow \beta\,\mathbf{Z}_{(t)}^{\top}\mathbf{Y}_{(t)}, \quad \{\textit{Likelihood component}\}$
10: $\quad \mathbf{W}^{(t)} \leftarrow \mathbf{R}^{(t)}\,(\mathbf{m}_{\mathrm{prior}} + \mathbf{m}_{\mathrm{data}})$
11: **end for**
12: **Output:** Final posterior parameters $\mathbf{W}^{(T)}$, $\mathbf{R}^{(T)}$

---

Using equation 6, we further introduce a precision matrix $\mathbf{P}_{\mathrm{new}}^{(t)}$, which integrates three synergistic components: the prior precision to retain knowledge from previous tasks, a Fisher-based regularizer to constrain essential parameters, and a data-driven term that facilitates adaptation to the current task:

$$\mathbf{P}_{\mathrm{new}}^{(t)} = \underbrace{\big(\mathbf{R}^{(t-1)}\big)^{-1}}_{\textbf{Prior Knowledge Confidence}} + \underbrace{\eta\,\lambda_{\mathrm{fisher}}}_{\textbf{Knowledge Preservation}} + \underbrace{\beta\,\mathbf{Z}_{(t)}^{\top}\mathbf{Z}_{(t)}}_{\textbf{Plasticity-Driven Adaptation}} \tag{7}$$

The posterior covariance $\mathbf{R}^{(t)}$ is obtained by inverting the total precision matrix (regularized by a small $\epsilon$):

$$\mathbf{R}^{(t)} = \big(\mathbf{P}_{\mathrm{new}}^{(t)} + \epsilon\mathbf{I}\big)^{-1} \tag{8}$$

Therefore, using equation 7 and equation 8, the classification layer weights ($\mathbf{W}^{(t)}$) are obtained as a precision-weighted combination of prior knowledge and task-specific evidence:

$$\mathbf{W}^{(t)} = \mathbf{R}^{(t)}\bigg[\Big(\big(\mathbf{R}^{(t-1)}\big)^{-1} + \eta\,\boldsymbol{\lambda}_{\mathrm{fisher}}\Big)\mathbf{W}^{(t-1)} + \beta\,\mathbf{Z}_{(t)}^{\top}\mathbf{Y}_{(t)}\bigg] \tag{9}$$

Here, the term $\big(\mathbf{R}^{(t-1)}\big)^{-1}\mathbf{W}^{(t-1)}$) captures the influence of past parameters weighted by their precision. In contrast, $\eta\lambda_{\mathrm{fisher}}\mathbf{W}^{(t-1)}$ penalizes deviations from important parameters identified via Fisher Information. The evidence term $\beta\mathbf{Z}_{(t)}^{\top}\mathbf{Y}_{(t)}$ introduces new task-specific information, enabling adaptation. The model balances stability and plasticity by adjusting $\eta$ and $\beta$, preserving critical past knowledge while adapting to new tasks. The updated precision matrix helps to reduce uncertainty and prior-informed confidence, facilitating smooth knowledge transfer and improving continual learning performance, as represented in Table 1. The pseudo-code of the proposed model is listed in Algorithm 1.

## 4 Experimental Setup

This section provides an explicit overview of the benchmark datasets, evaluation protocols, and implementation parameters adopted for validating the proposed method.

**Datasets.** We evaluate our method on three widely used benchmarks in continual learning: CIFAR-100 Krizhevsky et al. (2009), ImageNet-R (200 classes) Hendrycks et al. (2021), ImageNet-1000 (1000 classes) Krizhevsky et al. (2017) and Tiny-ImageNet (200 classes) Le & Yang (2015), covering diverse visual domains and varying task complexities. We employ the Si-Blurry protocol Moon et al. (2023), a challenging GCIL scenario designed to evaluate model robustness in the presence of indistinct task boundaries and complex

| Mem Size | Method | EFCIL | CIFAR-100 (%) | | | ImageNet-R (%) | | | Tiny-ImageNet (%) | | |
|---|---|---|---|---|---|---|---|---|---|---|---|
| | | | $\mathcal{A}_{\mathrm{AUC}}$ | $\mathcal{A}_{\mathrm{Avg}}$ | $\mathcal{A}_{\mathrm{Last}}$ | $\mathcal{A}_{\mathrm{AUC}}$ | $\mathcal{A}_{\mathrm{Avg}}$ | $\mathcal{A}_{\mathrm{Last}}$ | $\mathcal{A}_{\mathrm{AUC}}$ | $\mathcal{A}_{\mathrm{Avg}}$ | $\mathcal{A}_{\mathrm{Last}}$ |
| 2000 | EWC++ Kirkpatrick et al. (2017) | ✗ | $53.31_{\pm1.7}$ | $50.95_{\pm1.5}$ | $52.55_{\pm0.7}$ | $36.31_{\pm0.7}$ | $39.87_{\pm1.3}$ | $29.52_{\pm0.4}$ | $52.43_{\pm0.5}$ | $54.61_{\pm1.5}$ | $37.67_{\pm0.7}$ |
| | ER Rolnick et al. (2019) | ✗ | $56.17_{\pm1.8}$ | $53.80_{\pm1.4}$ | $55.60_{\pm0.6}$ | $39.31_{\pm0.7}$ | $43.03_{\pm1.1}$ | $32.09_{\pm0.4}$ | $55.69_{\pm0.4}$ | $57.87_{\pm1.4}$ | $41.10_{\pm0.5}$ |
| | RM Bang et al. (2021) | ✗ | $53.22_{\pm1.8}$ | $52.99_{\pm1.6}$ | $55.25_{\pm0.6}$ | $32.34_{\pm1.8}$ | $36.46_{\pm2.2}$ | $25.26_{\pm1.0}$ | $49.28_{\pm0.4}$ | $57.74_{\pm1.5}$ | $41.79_{\pm0.3}$ |
| | MVP-R Moon et al. (2023) | ✗ | $60.62_{\pm1.2}$ | $57.58_{\pm0.5}$ | $64.30_{\pm0.2}$ | $\underline{47.32_{\pm1.1}}$ | $\underline{50.16_{\pm0.9}}$ | $42.05_{\pm0.1}$ | $61.15_{\pm0.8}$ | $62.41_{\pm0.5}$ | $51.12_{\pm0.6}$ |
| | Online-LoRA Wei et al. (2025) | ✗ | $52.84_{\pm7.9}$ | $51.58_{\pm1.2}$ | $58.72_{\pm1.4}$ | $39.37_{\pm1.9}$ | $35.61_{\pm3.6}$ | $36.61_{\pm4.6}$ | $15.35_{\pm0.6}$ | $18.41_{\pm0.4}$ | $20.18_{\pm1.8}$ |
| | MISA KANG et al. (2025) | ✗ | $60.12_{\pm2.4}$ | $57.12_{\pm2.3}$ | $61.13_{\pm1.2}$ | $46.33_{\pm1.8}$ | $48.43_{\pm2.5}$ | $41.43_{\pm1.1}$ | $61.44_{\pm2.6}$ | $64.32_{\pm0.4}$ | $62.21_{\pm1.2}$ |
| 500 | EWC++ Kirkpatrick et al. (2017) | ✗ | $48.31_{\pm1.8}$ | $44.56_{\pm0.9}$ | $40.52_{\pm0.8}$ | $32.81_{\pm0.7}$ | $35.54_{\pm1.6}$ | $23.43_{\pm0.6}$ | $45.30_{\pm0.6}$ | $46.34_{\pm2.05}$ | $27.05_{\pm1.3}$ |
| | ER Rolnick et al. (2019) | ✗ | $51.5_{\pm1.9}$ | $48.03_{\pm0.8}$ | $44.09_{\pm0.8}$ | $35.96_{\pm0.7}$ | $39.01_{\pm1.5}$ | $26.14_{\pm0.4}$ | $48.95_{\pm0.5}$ | $50.44_{\pm1.7}$ | $29.97_{\pm0.7}$ |
| | RM Bang et al. (2021) | ✗ | $41.07_{\pm1.3}$ | $38.10_{\pm0.5}$ | $32.66_{\pm0.3}$ | $22.45_{\pm0.6}$ | $22.08_{\pm1.7}$ | $19.61_{\pm0.1}$ | $36.66_{\pm0.4}$ | $38.83_{\pm2.3}$ | $18.23_{\pm0.2}$ |
| | MVP-R Moon et al. (2023) | ✗ | $56.20_{\pm1.4}$ | $53.61_{\pm0.04}$ | $55.3_{\pm0.4}$ | $43.28_{\pm1.4}$ | $45.74_{\pm0.9}$ | $35.60_{\pm1.1}$ | $55.28_{\pm1.4}$ | $55.45_{\pm1.2}$ | $40.12_{\pm0.4}$ |
| | Online-LoRA Wei et al. (2025) | ✗ | $48.62_{\pm2.6}$ | $46.58_{\pm1.5}$ | $52.30_{\pm0.2}$ | $35.16_{\pm1.0}$ | $33.36_{\pm0.5}$ | $37.05_{\pm0.2}$ | $12.15_{\pm0.8}$ | $15.41_{\pm0.8}$ | $17.12_{\pm0.2}$ |
| | MISA KANG et al. (2025) | ✗ | $55.32_{\pm1.9}$ | $54.11_{\pm1.4}$ | $50.43_{\pm0.5}$ | $40.12_{\pm0.6}$ | $42.32_{\pm2.1}$ | $39.43_{\pm0.9}$ | $57.43_{\pm2.1}$ | $59.32_{\pm1.4}$ | $55.68_{\pm1.2}$ |
| 0 | SLDA Hayes & Kanan (2020) | ✓ | $53.00_{\pm3.8}$ | $50.09_{\pm2.7}$ | $61.79_{\pm3.8}$ | $33.11_{\pm3.1}$ | $33.78_{\pm1.7}$ | $39.02_{\pm1.3}$ | $49.17_{\pm4.4}$ | $47.93_{\pm4.4}$ | $53.13_{\pm2.2}$ |
| | L2P Wang et al. (2022b) | ✓ | $42.68_{\pm2.7}$ | $39.89_{\pm0.4}$ | $28.59_{\pm3.3}$ | $30.21_{\pm0.9}$ | $32.21_{\pm1.7}$ | $18.01_{\pm3.0}$ | $41.67_{\pm1.1}$ | $42.53_{\pm2.5}$ | $24.78_{\pm2.3}$ |
| | DualPrompt Wang et al. (2022a) | ✓ | $41.34_{\pm2.5}$ | $38.59_{\pm0.6}$ | $22.74_{\pm3.4}$ | $30.44_{\pm0.8}$ | $32.54_{\pm1.8}$ | $16.0_{\pm3.2}$ | $39.16_{\pm1.1}$ | $39.81_{\pm3.0}$ | $20.42_{\pm3.3}$ |
| | MVP Moon et al. (2023) | ✓ | $45.07_{\pm2.4}$ | $44.93_{\pm0.5}$ | $39.94_{\pm0.4}$ | $35.77_{\pm2.5}$ | $35.58_{\pm1.2}$ | $22.06_{\pm5.0}$ | $46.43_{\pm3.0}$ | $45.41_{\pm1.0}$ | $28.21_{\pm2.8}$ |
| | GACL Zhuang et al. (2024a) | ✓ | $57.99_{\pm2.4}$ | $56.24_{\pm3.1}$ | $70.31_{\pm0.6}$ | $41.68_{\pm0.7}$ | $47.30_{\pm0.8}$ | $42.21_{\pm0.1}$ | $63.14_{\pm0.6}$ | $69.32_{\pm0.8}$ | $62.68_{\pm0.8}$ |
| | MISA KANG et al. (2025) | ✓ | $48.11_{\pm2.4}$ | $31.12_{\pm3.3}$ | $42.19_{\pm1.5}$ | $28.43_{\pm0.7}$ | $44.21_{\pm0.8}$ | $41.12_{\pm0.1}$ | $59.24_{\pm0.6}$ | $67.32_{\pm0.8}$ | $60.68_{\pm1.2}$ |
| | AnaCP Momeni et al. (2025) | ✓ | $57.88_{\pm1.3}$ | $57.01_{\pm2.1}$ | $70.91_{\pm1.3}$ | $41.96_{\pm0.4}$ | $47.88_{\pm0.7}$ | $42.55_{\pm1.1}$ | $64.74_{\pm0.5}$ | $70.02_{\pm0.9}$ | $62.96_{\pm1.7}$ |
| | **AACL (ours)** | ✓ | $\underline{65.91_{\pm1.4}}$ | $\underline{62.21_{\pm1.1}}$ | $\underline{72.01_{\pm0.1}}$ | $44.78_{\pm0.7}$ | $49.70_{\pm1.2}$ | $\underline{44.01_{\pm0.2}}$ | $\underline{67.10_{\pm0.3}}$ | $\underline{72.02_{\pm0.5}}$ | $\underline{64.18_{\pm0.2}}$ |

Table 1: Evaluation of $\mathcal{A}_{\mathrm{AUC}}$, $\mathcal{A}_{\mathrm{Avg}}$, and $\mathcal{A}_{\mathrm{Last}}$ metrics for **AACL** in comparison with state-of-the-art methods under a five-task setting. All experiments are conducted using the DeiT-S/16 backbone. Each experiment is repeated five times, and results are reported as *mean ± std*. The underlined entries denote the best performance across all settings. To ensure consistency, we use identical hyperparameters and the same frozen backbone across all experiments under the *Si-Blurry* setting.

data stream dynamics. For evaluation purposes, we design a challenging streaming scenario with disjoint class ratio $r_D = 50\%$ and blurry sample ratio $r_B = 10\%$, which serves as a rigorous stress test for resistance to catastrophic forgetting. Explicit details of the Si-Blurry setting and other hyperparameters are given in the Appendix.

**Implementation Details.** To ensure fair and rigorous evaluation, we adopt a frozen DeiT-S/16 Touvron et al. (2021) as the backbone feature extractor. This model was pre-trained on a strategically curated 611-class ImageNet subset, ensuring no classes overlap with downstream evaluation datasets to maintain experimental integrity and prevent information leakage. We evaluate performance using memory buffer sizes of 500 and 2,000 samples for replay-based methods to assess scalability across different memory constraints. We adopt hyperparameter configurations consistent with those of GACL Zhuang et al. (2024a) to ensure reproducibility and provide fair comparisons. Specifically, the buffer layer dimensionality is fixed at 5000 across all tasks, and the regularization coefficient $\gamma$ is set to 100. We perform experiments on an NVIDIA L40S GPU using batch sizes of 64 for training and 128 for inference.

**Evaluation Metrics.** To assess the efficacy of our proposed methodology, we adopt three standard metrics, consistent with the evaluation protocol in Koh et al. (2022); Zhuang et al. (2024a). These include the area under the accuracy curve, average incremental accuracy, and last-task accuracy. These metrics provide comprehensive evaluation of CL performance. They measure inference capability, knowledge retention, and adaptability across sequential tasks.

1. **Area Under the Curve of Accuracy ($A_{\mathbf{AUC}}$).** This metric is designed to evaluate the anytime inference capability of continual learning models. it measures accuracy at regular intervals throughout training by computing the area under the accuracy-to-sample curve. It is computed as: $A_{\mathrm{AUC}} =$

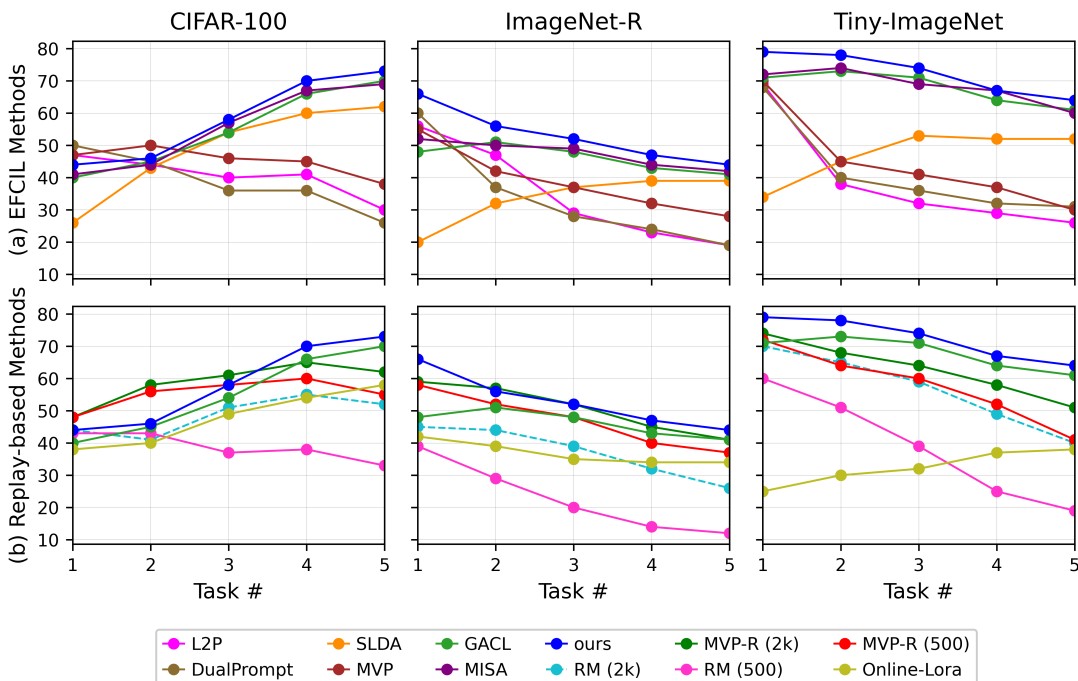

Figure 2: Task-wise accuracy ($A_t$) comparison of **AACL** with state-of-the-art EFCIL methods (top) and replay-based approaches (bottom) across benchmark datasets, evaluated over five incremental tasks.

$\sum_{i=1}^{T} f(i \cdot \Delta n) \cdot \Delta n$. where, $\Delta n$ is the evaluation interval, and $f(\cdot)$ denotes accuracy over training. A higher $A_{\text{AUC}}$ reflects more consistent performance throughout learning.

2. **Average Incremental Accuracy ($A_{\textbf{Avg}}$).** This metric evaluates the model's ability to retain knowledge across tasks. It is defined as: $A_{\text{Avg}} = \frac{1}{T+1} \sum_{t=1}^{T} A_t$. where, $A_t$ denotes the average accuracy on $D_{1:T}^{\text{test}}$ following task $T$.

3. **Last-Task Accuracy ($A_{\textbf{Last}}$).** This metric measures the model's final performance on the most recent task. It reflects the model's ability to adapt to new information without being hindered by previous learning.

4. **Forgetting ($\mathcal{F}_{\textbf{Last}}$)** Forgetting metric quantifies catastrophic forgetting by measuring the average accuracy drop between the best accuracy achieved for each task during training and its final accuracy after all tasks have been learned: $\mathcal{F} = \frac{1}{T-1} \sum_{i=1}^{T-1} \left( \max_{t \leq T} a_{t,i} - a_{T,i} \right)$.

## 5 Benchmark Comparison

Our comparative analysis includes various CL algorithms from distinct methodological categories: replay-based methods such as EWC++ Kirkpatrick et al. (2017), ER Rolnick et al. (2019), RM Bang et al. (2021), MVP-R Moon et al. (2023), Online-LoRA Wei et al. (2025), and exemplar-free methods including SLDA Hayes & Kanan (2020), L2P Wang et al. (2022b), DualPrompt Wang et al. (2022a), MVP Moon et al. (2023), GACL Zhuang et al. (2024a), MISA KANG et al. (2025), and AnaCP Momeni et al. (2025), currently the top-performing methods in the generalized continual learning domain.

**Comparison with EFCIL Methods.** Exemplar-free methods address the dual challenges of privacy preservation and catastrophic forgetting without relying on historical exemplars. Among these, AACL demonstrates consistent and significant improvements across all three benchmark datasets, as reported in Table 1. On CIFAR-100, a coarse-grained benchmark where frozen backbone features yield well-separated

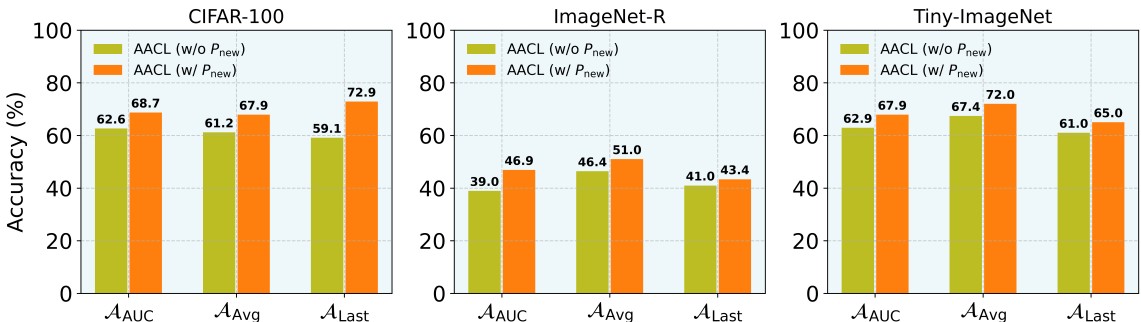

Figure 3: Ablation study evaluating the impact of incorporating *precision matrix* ($P_{\text{new}}$) in the proposed method across *CIFAR-100*, *ImageNet-R*, and *Tiny-ImageNet* datasets. Results are presented for three key metrics: $\mathcal{A}_{\text{AUC}}$, $\mathcal{A}_{\text{Avg}}$, and $\mathcal{A}_{\text{Last}}$. The inclusion of $P_{\text{new}}$ matrix leads to consistent performance improvements, underscoring its contribution to the overall learning efficacy.

| Buffer Layer Size | Time (s) | Memory (GB) | CIFAR-100 (%) | | | ImageNet-R (%) | | | Tiny-ImageNet (%) | | |
|---|---|---|---|---|---|---|---|---|---|---|---|
| | | | $\mathcal{A}_{\text{AUC}}$ | $\mathcal{A}_{\text{Avg}}$ | $\mathcal{A}_{\text{Last}}$ | $\mathcal{A}_{\text{AUC}}$ | $\mathcal{A}_{\text{Avg}}$ | $\mathcal{A}_{\text{Last}}$ | $\mathcal{A}_{\text{AUC}}$ | $\mathcal{A}_{\text{Avg}}$ | $\mathcal{A}_{\text{Last}}$ |
| 1000 | 2.91 | 0.60 | 64.02 | 62.79 | 66.66 | 40.61 | 47.33 | 38.62 | 60.70 | 67.44 | 58.52 |
| 3000 | 2.99 | 0.66 | 66.98 | 66.11 | 70.37 | 43.46 | 48.79 | 42.40 | 65.25 | 70.93 | 62.56 |
| **5000** | 3.13 | 0.79 | **68.72** | 67.88 | **72.87** | **46.89** | 50.99 | **44.35** | **67.88** | **72.10** | **65.04** |
| 7000 | 3.22 | 0.97 | 68.71 | **67.89** | 72.85 | 46.88 | **60.01** | 44.34 | **67.88** | 72.08 | 65.02 |
| 10000 | 3.40 | 1.36 | 68.72 | 67.88 | 71.69 | 46.71 | 49.47 | 43.31 | 67.81 | 72.04 | 64.97 |

Table 2: We evaluate model behavior under varying linear buffer sizes across multiple datasets. Accuracy improves steadily to an optimal buffer size, while time and memory overheads increase marginally when buffer size doubles, demonstrating our analytical layer's efficiency.

| Task | 1 | 2 | 3 | 4 | 5 | 6 | 7 | 8 | 9 | 10 |
|---|---|---|---|---|---|---|---|---|---|---|
| #Classes | 212 | 363 | 397 | 495 | 597 | 676 | 733 | 824 | 933 | 1000 |
| Accuracy (%) | 78.0 | 85.7 | 87.0 | 86.9 | 88.3 | 87.5 | 87.5 | 86.6 | 85.9 | 85.4 |
| **Performance (AACL) :** $\mathcal{A}_{\text{AUC}}$ : 83.98 | | $\mathcal{A}_{\text{Avg}}$ : 85.92 | | | $\mathcal{A}_{\text{Last}}$ : 85.49 | | | $\mathcal{F}_{\text{Last}}$: 5.31 | | |

Table 3: Scalability analysis on **ImageNet-1000** with DeiT-S/16 under a 10-task continual learning setting, reporting $\mathcal{A}_{\text{AUC}}$, $\mathcal{A}_{\text{Avg}}$, $\mathcal{A}_{\text{Last}}$, and $\mathcal{F}_{\text{Last}}$ (forgetting) as percentages over the best of five runs.

class embeddings, AACL outperforms the previous best-performing EFCIL method, GACL, by notable margins of 8.0%, 5.9%, and 1.70% in $\mathcal{A}_{\text{AUC}}$, $\mathcal{A}_{\text{Avg}}$, and $\mathcal{A}_{\text{Last}}$, respectively. This strong performance is attributed to the feature separability of the frozen backbone, where coarse-grained classes produce a stronger $\mathbf{Z}^{\top}\mathbf{Z}$, reinforcing the analytic update. Similar gains are observed on Tiny-ImageNet, where AACL surpasses GACL by 3.0%, 2.40%, and 1.50% across the same metrics. On the more challenging ImageNet-R dataset, which exhibits high visual diversity and distribution-shifted fine-grained categories, AACL still outperforms GACL with gains of 4.0%, 2.70%, and 1.5%. The comparatively lower performance on ImageNet-R reflects the inherent difficulty of fine-grained discrimination under a frozen backbone, where overlapping embeddings challenge closed-form separation. Nevertheless, the Fisher-based regularization $\boldsymbol{\lambda}_{(t)}$ effectively mitigates this by retaining discriminative parameters across tasks, ensuring robust performance regardless of dataset granularity. Furthermore, as the task index $T$ increases, AACL retains stable performance in contrast to the degradation observed in baselines such as SLDA, L2P, and MVP, affirming its effectiveness in preserving previously acquired knowledge without exemplar replay.

**Comparison with Replay-Based Methods.** Replay-based methods consistently demonstrate strong performance in CIL, primarily due to their ability to access stored exemplars. Their performance often scales with memory size, but at the cost of increased storage and potential privacy concerns. In contrast, despite

| Method | ImageNet-R (%) | | | | Tiny-ImageNet (%) | | | |
|---|---|---|---|---|---|---|---|---|
| | $\mathcal{A}_{\text{AUC}}$ | $\mathcal{A}_{\text{Avg}}$ | $\mathcal{A}_{\text{Last}}$ | $\mathcal{F}_{\text{Last}}$ | $\mathcal{A}_{\text{AUC}}$ | $\mathcal{A}_{\text{Avg}}$ | $\mathcal{A}_{\text{Last}}$ | $\mathcal{F}_{\text{Last}}$ |
| AACL (Baseline) | 46.1 | 51.1 | 43.3 | 6.6 | 66.7 | 72.1 | 64.8 | 9.2 |
| AACL + LoRA (Joint Training) | 27.2 | 29.9 | 30.4 | 27.6 | 33.1 | 31.2 | 32.0 | 29.4 |
| AACL + LoRA (Frozen) | 48.3 | 53.8 | 47.1 | 4.2 | 69.5 | 75.7 | 66.8 | 7.4 |

Table 4: Performance comparison of AACL and LoRA variants on DeiT-S/16 across a 5-task continual learning setting at rank 10.

| Method | CIFAR-100 (%) | | | | ImageNet-R (%) | | | | Tiny-ImageNet (%) | | | |
|---|---|---|---|---|---|---|---|---|---|---|---|---|
| | $\mathcal{A}_{\text{AUC}}$ | $\mathcal{A}_{\text{Avg}}$ | $\mathcal{A}_{\text{Last}}$ | $T_{\text{Time}}$ | $\mathcal{A}_{\text{AUC}}$ | $\mathcal{A}_{\text{Avg}}$ | $\mathcal{A}_{\text{Last}}$ | $T_{\text{Time}}$ | $\mathcal{A}_{\text{AUC}}$ | $\mathcal{A}_{\text{Avg}}$ | $\mathcal{A}_{\text{Last}}$ | $T_{\text{Time}}$ |
| **AACL (wo/Woodbury)** | 68.16 | 67.36 | 71.82 | 32.49 | 46.07 | 51.08 | 43.43 | 16.15 | 66.79 | 72.01 | 64.08 | 64.00 |
| **AACL (w/Woodbury)** | 68.64 | 67.36 | 71.82 | **17.41** | 46.07 | 51.07 | 43.44 | **8.58** | 66.79 | 71.99 | 64.08 | **34.39** |

Table 5: Performance and efficiency comparison of AACL with and without Woodbury identity optimization across three benchmarks under the *Si-Blurry* protocol. We report $\mathcal{A}_{\text{AUC}}$, $\mathcal{A}_{\text{Avg}}$, $\mathcal{A}_{\text{Last}}$, and total training time $T_{\text{Time}}$ (in minutes). AACL (w/Woodbury) employs the Woodbury matrix identity to reduce computational complexity, achieving substantial speedup while maintaining comparable accuracy. Results represent the best performance across five independent runs (5 Task) with identical hyperparameters and network architecture.

operating without access to historical data, AACL consistently outperforms several strong replay-based baselines. On CIFAR-100, AACL outperforms the strongest replay-based competitor, MVP-R (with a memory buffer of 2000 samples), by 5.29%, 4.63%, and 7.71% in $\mathcal{A}_{\text{AUC}}$, $\mathcal{A}_{\text{Avg}}$, and $\mathcal{A}_{\text{Last}}$, respectively. A similar pattern is observed on Tiny-ImageNet, where AACL achieves improvements of 5.95%, 9.61%, and 13.06%. Even on ImageNet-R, where MVP-R performs competitively, AACL shows marginal decreases of 2.38% in $\mathcal{A}_{\text{AUC}}$ and 0.66% in $\mathcal{A}_{\text{Avg}}$, while maintaining a positive gain of 1.96% in $\mathcal{A}_{\text{Last}}$. These results highlight AACL's ability to consistently outperform memory-based replay strategies while avoiding the storage and privacy challenges associated with exemplar retention. While replay methods suffer from gradient-induced performance decay, AACL maintains consistently stable accuracy.

**Task-wise Accuracy Comparison.** Figure 2 presents task-wise accuracy $\mathcal{A}_t$ comparisons between AACL and state-of-the-art EFCIL methods (top) and exemplar-replay approaches (bottom) on CIFAR-100, Tiny-ImageNet, and ImageNet-R, demonstrating AACL's consistent superiority across all baseline methods. On CIFAR-100, AACL maintains a steady increase in accuracy from the first to the last task (44% to 73%), outperforming all EFCIL methods such as GACL and MVP, which suffer from performance drop-off as task increases. Similar patterns are observed on Tiny-ImageNet, where AACL outpaces the closest EFCIL method by over 6% on the final task. On ImageNet-R, AACL provides strong generalization, maintaining high $A_t$ values even on later tasks where competing methods such as SLDA or MVP deteriorate. Notably, AACL surpasses the best replay-based methods such as MVP-R (2000 memory size), demonstrating competitive performance while preserving privacy.

## 6 Ablation Study

We present an ablation study evaluating the impact of incorporating the precision matrix $\mathbf{P}_{\text{new}}$ within the AACL framework across all three datasets in Figure 3. The results demonstrate that incorporating $P_{\text{new}}$ consistently enhances performance across all evaluated configurations. The inclusion of $P_{\text{new}}$ yields significant performance enhancements across benchmark datasets. On CIFAR-100, we observe improvements of 5.27% and 12.8% for $\mathcal{A}_{\text{AUC}}$ and $\mathcal{A}_{\text{Last}}$, respectively, while ImageNet-R shows gains of 7.9% and 2.4% for the corresponding metrics. These substantial improvements underscore the fundamental importance of precision matrix integration in achieving AACL's state-of-the-art performance. Table 2 illustrates the influence of buffer size on AACL's performance across all datasets. The results indicate a clear and consistent trend: increasing the buffer size enhances knowledge retention and task adaptation. Performance steadily

| Model | ImageNet-R (%) | | | | Tiny-ImageNet (%) | | | |
|---|---|---|---|---|---|---|---|---|
| | $\mathcal{A}_{\text{AUC}}$ | $\mathcal{A}_{\text{Avg}}$ | $\mathcal{A}_{\text{Last}}$ | $\mathcal{F}_{\text{Last}}$ | $\mathcal{A}_{\text{AUC}}$ | $\mathcal{A}_{\text{Avg}}$ | $\mathcal{A}_{\text{Last}}$ | $\mathcal{F}_{\text{Last}}$ |
| ResNet18 wo/(G+P) | 51.7 | 57.0 | 48.8 | 7.0 | 66.1 | 63.3 | 48.8 | 8.2 |
| ResNet18 w/(G+P) | 49.8 | 54.7 | 48.5 | 7.2 | 64.9 | 63.2 | 48.5 | 8.5 |
| ViT-B/16 wo/(G+P) | 59.1 | 64.9 | 60.3 | 4.4 | 83.4 | 82.2 | 60.3 | 4.4 |
| ViT-B/16 w/(G+P) | 57.6 | 63.1 | 60.2 | 4.8 | 83.1 | 82.0 | 60.0 | 4.6 |

Table 6: Performance evaluation under the Gaussian and Periodic input data stream (G+P) Koh et al. (2023) scenario using ResNet18 and ViT-B/16 backbones on ImageNet-R and Tiny-ImageNet datasets.

| Dataset | Task#10 | | | | Task#20 | | | |
|---|---|---|---|---|---|---|---|---|
| | $\mathcal{A}_{\text{AUC}}$ | $\mathcal{A}_{\text{Avg}}$ | $\mathcal{A}_{\text{Last}}$ | $\mathcal{F}_{\text{Last}}$ | $\mathcal{A}_{\text{AUC}}$ | $\mathcal{A}_{\text{Avg}}$ | $\mathcal{A}_{\text{Last}}$ | $\mathcal{F}_{\text{Last}}$ |
| CIFAR-100 | 60.9 | 61.8 | 72.2 | 7.6 | 52.0 | 54.7 | 72.1 | 7.9 |
| ImageNet-R | 47.9 | 49.3 | 43.7 | 8.4 | 45.0 | 43.6 | 43.5 | 9.7 |
| Tiny-ImageNet | 66.7 | 70.0 | 64.4 | 8.9 | 66.0 | 67.4 | 64.2 | 9.8 |

Table 7: Scalability evaluation of AACL using DeiT-S/16 as the backbone over extended continual learning sequences, reporting $\mathcal{A}_{\text{AUC}}$, $\mathcal{A}_{\text{Avg}}$, $\mathcal{A}_{\text{Last}}$, and $\mathcal{F}_{\text{Last}}$ (forgetting) as percentages across increasing task lengths of $T \in \{10, 20\}$ on CIFAR-100, ImageNet-R, and Tiny-ImageNet datasets.

improves as the buffer expands from 1000 to 5000, with the most notable gains occurring in the shift from smaller to larger buffers. On CIFAR-100, a buffer size of 5000 delivers the best outcomes across all metrics, achieving gains of 4.7% ($\mathcal{A}_{\text{AUC}}$), 5.09% ($\mathcal{A}_{\text{Avg}}$), and 6.21% ($\mathcal{A}_{\text{Last}}$) compared to the smallest buffer setting. We also tested increasing the buffer size to 10k, which showed only marginal improvements at the decimal level. Hence, the 5000 buffer size is optimal to achieve state-of-the-art results, confirming that adequate buffer capacity is crucial for AACL's effectiveness in CL scenarios.

**Backbone Fine-Tuning with LoRA.** We conduct an ablation study that jointly trains LoRA adapters using DeiT-S/16 with the analytic classifier update. This configuration leads to significant training instability and drastically degraded accuracy compared to the frozen-backbone setting. This occurs because AACL's closed-form update follows the weight-invariant property, where the recursively accumulated precision matrix $\mathbf{P}^{(t-1)}$ and prior $\mathbf{W}^{(t-1)}$ capture statistics bound to the backbone's embedding distribution; LoRA updates shift this distribution across tasks, invalidating the prior statistics and breaking the recursive equivalence. Furthermore, for LoRA integration with AACL, we adopt a two-stage strategy: DeiT-S/16 is first fine-tuned with LoRA adapters and subsequently frozen prior to the analytical updates. This preserves the weight-invariant property while leveraging LoRA's richer feature representations, resulting in an average accuracy improvement of 3%, as reported in Table 4.

**Scalability Evaluation of AACL.** Table 3 presents the scalability evaluation of AACL on the large-scale ImageNet-1000 benchmark, where the model incrementally learns 1000 classes over 10 sequential tasks. AACL maintains strong and stable performance throughout the sequence, with task accuracies consistently remaining in the 78–88% range despite the increasing number of learned classes. The aggregate metrics further confirm its robustness, achieving an $\mathcal{A}_{\text{Avg}}$ of 85.9% and an $\mathcal{A}_{\text{Last}}$ of 85.4%, while maintaining a low forgetting rate of just 5.3%. These results demonstrate that AACL scales effectively to high-capacity settings, preserving previously learned knowledge while continuing to acquire new classes with minimal degradation. Table 6 demonstrates that AACL maintains strong generalizability across complex stream settings with minimal accuracy degradation. ViT-B/16 consistently outperforms ResNet18 under the Gaussian and Periodic (G+P) scenario (Koh et al., 2023) across all metrics. Notably, the accuracy drop when introducing gaussian perturbations remains marginal for ViT-B/16, with $\mathcal{A}_{\text{AUC}}$ declining from 59.1 to 57.6, whereas ResNet18 exhibits slightly larger degradation. Furthermore, $\mathcal{F}_{\text{Last}}$ confirms the stability of AACL, as ViT-B/16 retains consistently low forgetting scores ranging from 4.4 to 4.8 even under perturbed stream conditions, highlighting its robustness to non-stationary data distributions. Furthermore, Table 7 evaluates the scalability of AACL under task sequence lengths of 10 and 20. As the number of tasks increases from 10 to 20, AACL exhibits

| Dataset | Model Type | $\mathcal{A}_{\mathrm{AUC}}$ (%) | $\mathcal{A}_{\mathrm{Avg}}$ (%) | $\mathcal{A}_{\mathrm{Last}}$ (%) |
|---|---|---|---|---|
| CIFAR-100 | DINOv2 | **75.02** | **79.72** | **88.37** |
| | ViT-B/16 | 69.81 | 69.69 | 79.75 |
| | DeiT-S/16 | 68.12 | 67.38 | 72.17 |
| | ResNet34 | 65.63 | 64.41 | 68.31 |
| | ResNet50 | 67.70 | 66.74 | 71.02 |
| | EfficientNet | 67.28 | 66.89 | 71.53 |
| ImageNet-R | DINOv2 | **81.17** | **84.45** | **80.45** |
| | ViT-B/16 | 59.12 | 64.92 | 60.34 |
| | DeiT-S/16 | 44.78 | 50.29 | 44.55 |
| | ResNet34 | 55.05 | 60.05 | 52.12 |
| | ResNet50 | 58.19 | 62.89 | 55.45 |
| | EfficientNet | 50.12 | 55.77 | 49.48 |
| Tiny-ImageNet | DINOv2 | **87.69** | **89.40** | **87.23** |
| | ViT-B/16 | 83.46 | 86.32 | 82.21 |
| | DeiT-S/16 | 69.78 | 72.52 | 64.24 |
| | ResNet34 | 73.11 | 77.31 | 71.33 |
| | ResNet50 | 76.62 | 80.69 | 75.08 |
| | EfficientNet | 74.51 | 79.54 | 73.45 |

Table 8: Architectural generalizability of AACL across diverse backbone networks on three benchmark datasets. Underlined values indicate the highest performance among all configurations. Results represent the best performance from five independent runs.

graceful degradation, with $\mathcal{A}_{\mathrm{Last}}$ remaining largely stable across all datasets, most notably on CIFAR-100 and Tiny-ImageNet. The $\mathcal{F}_{\mathrm{Last}}$ scores increase only marginally with longer task sequences, confirming that AACL mitigates catastrophic forgetting even under more demanding continual learning settings. Overall, these results demonstrate that AACL scales reliably to longer task sequences without significant loss in discriminative performance across diverse dataset complexities.

**AACL Closed-Form vs. Woodbury Updates.** To assess the impact of computational efficiency on learning dynamics, we evaluate AACL under two formulations as shown in Table 5: the standard closed-form Bayesian update (AACL wo/Woodbury) and the Woodbury-optimized variant (AACL w/Woodbury). The standard formulation requires $O(D^3)$ matrix inversion for the precision matrix update, resulting in higher training costs, whereas the Woodbury identity reduces this complexity, yielding substantially faster training without affecting inference-time performance. Notably, both variants achieve nearly identical accuracy across all three benchmarks, demonstrating that the Woodbury optimization preserves AACL's full representational capacity while significantly reducing computational overhead. The marginal differences in $\mathcal{A}_{\mathrm{AUC}}$, $\mathcal{A}_{\mathrm{Avg}}$, and $\mathcal{A}_{\mathrm{Last}}$ metrics (within 0.5%) confirm that the mathematical reformulation maintains exact equivalence in practice. These results validate the Woodbury-accelerated update as both a practical and efficient solution for large-scale continual learning scenarios where training efficiency is critical.

**Backbone-Agnostic Performance Evaluation.** To evaluate the architectural generalizability of our approach, we conduct experiments across multiple backbone architectures, including DINOv2, DeiT-S/16, ViT-B/16, ResNet-34, ResNet-50, and EfficientNet-B0. This analysis demonstrates that AACL's effectiveness is independent of specific architectural choices. The results presented in Table 8 confirm consistent performance improvements across all evaluated architectures on all three datasets. The analysis shows that deep complex architectures, particularly DINOv2 and ViT-B/16, achieve superior performance across all metrics compared to compact architectures like ResNet-18. This superior performance is mainly due to the enhanced representational capacity of deeper architectures, which supports the extraction of more nuanced and discriminative features. These findings validate the architecture-agnostic nature of our approach, confirming its broad applicability across diverse model families. **Additional ablation studies are provided in the Appendix, including: (i) complexity verification with Woodbury updates, which reduce the complexity from $O(D^3)$ to $O(D^2)$, and (ii) hyperparameter sensitivity analysis**.

## 7 Conclusion and Future Work

In this work, we present AACL, a novel Exemplar-free Analytic Adaptation framework for continual learning (CL) in task-agnostic environments with blurry task boundaries. AACL performs CL without storing past data or relying on task labels by maintaining a full-covariance Gaussian posterior over the classifier weights. The core of AACL is a closed-form Bayesian update rule for the posterior precision matrix that integrates: (1) prior knowledge from past tasks, (2) Fisher-guided weight protection for critical features, and (3) new evidence from the current task. Consequently, AACL effectively balances stability and plasticity while avoiding the memory and privacy drawbacks of experience replay. We conduct comprehensive experiments under the challenging Si-Blurry CL protocol, evaluating AACL against state-of-the-art exemplar-free and replay-based methods on three benchmark datasets. Our results demonstrate significant performance improvements, with AACL achieving $\mathcal{A}_{\text{AUC}}$ gains of 8%, 3%, and 4% on CIFAR-100, ImageNet-R, and Tiny-ImageNet, respectively. These findings underscore AACL's robust performance in handling complex data streams and demonstrate its potential as an efficient and scalable framework for practical CL deployments.

In future work, we plan to extend AACL to multimodal settings by integrating vision-language backbones such as CLIP, broadening its applicability to heterogeneous data streams comprising both visual and textual modalities.

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

# A  Appendix

## A.1  Reproducibility and Code Availability

Code is available at https://anonymous.4open.science/r/Continual-Learning-6BFB/README.md

## A.2  Improving AACL Efficiency via Woodbury Updates

The precision matrix update combines three components:

$$\mathbf{P}_{\text{new}}^{(t)} = \underbrace{\mathbf{P}^{(t-1)}}_{\text{Prior Precision}} + \underbrace{\eta\mathbf{\Lambda}}_{\text{Fisher Regularization}} + \underbrace{\beta\mathbf{X}_{(t)}^{\top}\mathbf{X}_{(t)}}_{\text{Data Precision}} \tag{10}$$

where:

- $\mathbf{P}^{(t-1)} \in \mathbb{R}^{D \times D}$ is the prior precision matrix.

- $\mathbf{\Lambda} = \text{diag}(\lambda_1, \dots, \lambda_D)$ is the Fisher information matrix.

- $\mathbf{X}_{(t)} \in \mathbb{R}^{N \times D}$ is the batch of input features.

- $\eta, \beta > 0$ are hyperparameters.

The covariance matrix is obtained by inverting the precision matrix, which can be defined as:

$$\mathbf{R}^{(t)} = \left(\mathbf{P}_{\text{new}}^{(t)}\right)^{-1} \tag{11}$$

The weight update of the final layer then follows:

$$\mathbf{W}^{(t)} = \mathbf{R}^{(t)} \left[(\mathbf{P}^{(t-1)} + \eta\mathbf{\Lambda})\mathbf{W}^{(t-1)} + \beta\mathbf{X}_{(t)}^{\top}\mathbf{Y}_{(t)}\right] \tag{12}$$

where $\mathbf{Y}_{(t)} \in \mathbb{R}^{N \times K}$ represents the target outputs and $\mathbf{W}^{(t)} \in \mathbb{R}^{D \times K}$ are the final layer weights. The matrix inversion operation in Eq equation 11 represents the computational bottleneck, requiring $O(D^3)$ operations. We address this bottleneck by applying the Woodbury Matrix Identity with the Sherman-Morrison formula, which reduces the computational complexity to $O(D^2)$.

**Woodbury Matrix Identity.** The Woodbury matrix identity, also known as the matrix inversion lemma, provides a formula for the inverse of a matrix after a low-rank perturbation.

Let $\mathbf{A} \in \mathbb{R}^{D \times D}$ be an invertible matrix, $\mathbf{U} \in \mathbb{R}^{D \times k}$, $\mathbf{C} \in \mathbb{R}^{k \times k}$ be invertible, and $\mathbf{V} \in \mathbb{R}^{k \times D}$, then:

$$(\mathbf{A} + \mathbf{U}\mathbf{C}\mathbf{V})^{-1} = \mathbf{A}^{-1} - \mathbf{A}^{-1}\mathbf{U}(\mathbf{C}^{-1} + \mathbf{V}\mathbf{A}^{-1}\mathbf{U})^{-1}\mathbf{V}\mathbf{A}^{-1} \tag{13}$$

Let $\mathbf{C} = \mathbf{I}$ and $\mathbf{V} = \mathbf{U}^T$ in equation 13. Therefore, we can rewrite equation 13 as:

$$(\mathbf{A} + \mathbf{U}\mathbf{U}^T)^{-1} = \mathbf{A}^{-1} - \mathbf{A}^{-1}\mathbf{U}(\mathbf{I} + \mathbf{U}^T\mathbf{A}^{-1}\mathbf{U})^{-1}\mathbf{U}^T\mathbf{A}^{-1} \tag{14}$$

When $k \ll D$, the computational cost of inverting the $k \times k$ matrix $(\mathbf{I} + \mathbf{U}^T\mathbf{A}^{-1}\mathbf{U})$ is computationally far less expensive than inverting the original $D \times D$ matrix. To achieve this efficiency, we exploit the diagonal structure of $\mathbf{\Lambda}$ and apply sequential updates.

$$\mathbf{P}_{\text{new}}^{(t)} = \underbrace{(\mathbf{P}^{(t-1)} + \eta\mathbf{\Lambda})}_{\mathbf{A}} + \beta\mathbf{X}_{(t)}^{\top}\mathbf{X}_{(t)} \tag{15}$$

The data-dependent term in the precision update can be factorized to low-rank structure:

$$\beta\,\mathbf{X}_{(t)}^{\top}\mathbf{X}_{(t)} = \left(\sqrt{\beta}\,\mathbf{X}_{(t)}^{\top}\right)\left(\sqrt{\beta}\,\mathbf{X}_{(t)}^{\top}\right)^{\top} \tag{16}$$

$$\mathbf{A} = \mathbf{P}^{(t-1)} + \eta \mathbf{\Lambda}, \quad \mathbf{U} = \sqrt{\beta}\, \mathbf{X}_{(t)}^{\top} \tag{17}$$

where $\mathbf{P}^{(t-1)} = \left(\mathbf{R}^{(t-1)}\right)^{-1}$.

Consequently, the precision matrix update in equation 15 can be expressed as

$$\mathbf{P}_{\text{new}}^{(t)} = \mathbf{A} + \mathbf{U}\mathbf{U}^{\top} \tag{18}$$

Applying the Woodbury matrix identity from Eq. equation 14 to equation 18, the covariance matrix can be computed as:

$$\mathbf{R}^{(t)} = \left(\mathbf{P}_{\text{new}}^{(t)}\right)^{-1}$$
$$= \mathbf{A}^{-1} - \mathbf{A}^{-1}\mathbf{U}\left(\mathbf{I} + \mathbf{U}^{\top}\mathbf{A}^{-1}\mathbf{U}\right)^{-1}\mathbf{U}^{\top}\mathbf{A}^{-1} \tag{19}$$

Direct inversion of $\mathbf{A}$ remains computationally expensive for large $D$. However, the diagonal structure of $\mathbf{\Lambda}$ allows an efficient approximation using a diagonal variant of the Sherman–Morrison identity Ma et al. (2025). Therefore, we approximate $\mathbf{A}^{-1}$ as

$$\mathbf{A}^{-1} = \left(\mathbf{R}_{\text{prev}}^{-1} + \eta\mathbf{\Lambda}\right)^{-1}$$
$$\approx \mathbf{R}^{(t-1)} - \mathbf{R}^{(t-1)}\operatorname{diag}\left(\frac{\eta\lambda_i}{1 + \eta\lambda_i \mathbf{R}_{ii}^{(t-1)}}\right)\mathbf{R}^{(t-1)} \tag{20}$$

which reduces the computational cost from $O(D^3)$ to $O(D^2)$ per update while preserving the diagonal correction induced by the Fisher regularization. Given the updated covariance $\mathbf{R}^{(t)}$, the classifier weights are obtained as:

$$\mathbf{W}^{(t)} = \mathbf{R}^{(t)}\left[\mathbf{P}^{(t-1)}\mathbf{W}^{(t-1)} + \eta\mathbf{\Lambda}\mathbf{W}^{(t-1)} + \beta\mathbf{X}_{(t)}^{\top}\mathbf{Y}_{(t)}\right] \tag{21}$$

where $\mathbf{Y}_{(t)} \in \mathbb{R}^{N \times K}$ denotes the target outputs and $\mathbf{P}^{(t-1)} = (\mathbf{R}^{(t-1)})^{-1}$ is the prior precision matrix. To avoid computing $\mathbf{P}^{(t-1)} = (\mathbf{R}^{(t-1)})^{-1}$, which costs $O(D^3)$, we solve the linear system:

$$\mathbf{R}^{(t-1)}\mathbf{x} = \mathbf{W}^{(t-1)} \tag{22}$$

Here, $\mathbf{x}$ is defined as the solution to the linear system $\mathbf{R}^{(t-1)}\mathbf{x} = \mathbf{W}^{(t-1)}$, which is equivalent to $\mathbf{x} = \mathbf{P}^{(t-1)}\mathbf{W}^{(t-1)}$. Using the Cholesky decomposition Krishnamoorthy & Menon (2013), we factor $\mathbf{R}^{(t-1)} = \mathbf{L}\mathbf{L}^{\top}$. The linear system is then solved by forward substitution on $\mathbf{L}\mathbf{z} = \mathbf{W}^{(t-1)}$ followed by backward substitution on $\mathbf{L}^{\top}\mathbf{x} = \mathbf{z}$. Substituting $\mathbf{x}$ into Eq. equation 21, the final weight update becomes:

$$\mathbf{W}^{(t)} = \mathbf{R}^{(t)}\left(\mathbf{x} + \eta\mathbf{\Lambda}\mathbf{W}^{(t-1)} + \beta\mathbf{X}_{(t)}^{\top}\mathbf{Y}_{(t)}\right) \tag{23}$$

### A.3 Additional Ablation Analysis

In this section, we present a comprehensive ablation study to evaluate the robustness and generalizability of the proposed AACL method. Our analysis focuses on two critical aspects: the architectural generalizability of the method across diverse backbone networks and a sensitivity analysis of key hyperparameters that govern its performance. We further compare AACL with MISA using the same backbone on which MISA was originally proposed, to verify AACL's generalizability to new architectures, as shown in Table 9. AACL consistently outperforms MISA on both datasets under identical Si-Blurry settings with the ViT-B/16 backbone. On ImageNet-R, AACL achieves substantial improvements across all accuracy metrics while reducing forgetting by more than $4\times$ compared to MISA. Similar gains are observed on Tiny-ImageNet, where AACL demonstrates higher stability and stronger final-task performance. In addition to accuracy improvements, AACL offers significant efficiency advantages, including dramatically faster inference and lower peak GPU memory consumption. Overall, these results highlight AACL as a scalable, robust, and computation-efficient solution suitable for real-world continual learning deployments.

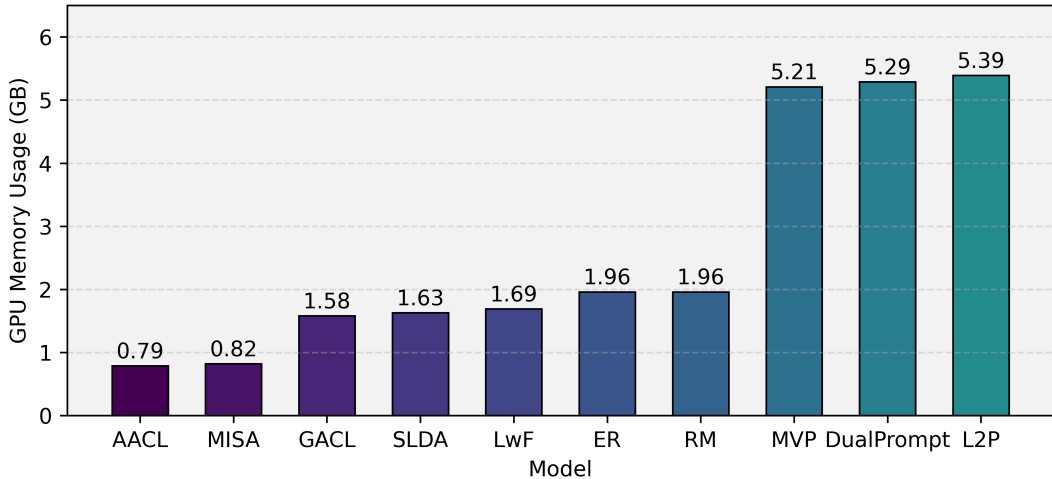

Figure 4: Comparison with baseline methods in terms of GPU memory consumption (GB) at a batch size of 64, where replay-based methods use 2000 exemplars with the DeiT-S/16 backbone.

| Dataset | Method | $\mathcal{A}_{\mathbf{AUC}}$ (%) | $\mathcal{A}_{\mathbf{Avg}}$ (%) | $\mathcal{A}_{\mathbf{Last}}$ (%) | $\mathcal{F}_{\mathbf{Last}}$ (%) |
|---|---|---|---|---|---|
| **ImageNet-R** | MISA | 51.4 | 55.5 | 43.2 | 17.1 |
| | **AACL** | **59.1** | **64.9** | **60.3** | **4.2** |
| **Tiny-ImageNet** | MISA | 80.8 | 82.7 | 75.8 | 12.0 |
| | **AACL** | **83.4** | **86.3** | **82.2** | **4.4** |
| MISA Inference Time (ms) | | 404.4 | | | |
| **AACL Inference Time (ms)** | | **3.1** | | | |
| MISA Peak GPU Memory (GB) | | 1.73 | | | |
| **AACL Peak GPU Memory (GB)** | | **1.50** | | | |

Table 9: Comparison of AACL with the strong baseline MISA under identical Si-Blurry settings using the ViT-B/16 Dosovitskiy (2020) backbone. Since MISA is originally proposed with **ViT-B/16**, we adopt the same architecture for a fair comparison. Results are reported as percentages and correspond to the best performance across five independent runs with different random seeds. We also report inference time and peak GPU memory usage, highlighting the substantial efficiency gains achieved by AACL.

**Memory Consumption (GPU).** We evaluate peak GPU memory efficiency using PyTorch on an NVIDIA L40 GPU server with batch sizes of 64 for training and 128 for inference. Figure 4 presents the inference-time memory footprint comparison across all methods. AACL achieves the lowest memory footprint (0.79 GB) among all evaluated methods, representing a 3.7% reduction compared to the latest approach, MISA (0.82 GB). When compared to other exemplar-free and replay-based methods, AACL demonstrates significant memory reduction. This efficiency results from eliminating exemplar storage and backpropagation overhead through replay-free and gradient-free optimization.

**Impact of Hyperparameters $\beta$ and $\eta$ on AACL Learning Performance.** Based on the empirical results presented in Table 10, the hyperparameters $\eta$ and $\beta$ play crucial roles in the proposed AACL method $(\mathbf{P}_{new}^{(t)} \leftarrow \left(\mathbf{R}^{(t-1)}\right)^{-1} + \eta\boldsymbol{\lambda}_t + \beta\mathbf{Z}_t^{\top}\mathbf{Z}_t)$. The parameter $\eta$ governs the influence of prior knowledge through the Fisher information, while $\beta$ controls the contribution of current task data during parameter updates. Our analysis across all datasets shows that lower values of $\eta$ (e.g., 20) consistently produce superior performance across all metrics ($\mathcal{A}_{\mathrm{AUC}}$, $\mathcal{A}_{\mathrm{Avg}}$, and $\mathcal{A}_{\mathrm{Last}}$) demonstrating that reduced reliance on Fisher information-based regularization facilitates more effective adaptation to new tasks. In contrast, while moderate values of $\beta$ enhance learning from current task data, increasing $\beta$ beyond an optimal threshold results in performance degradation. This degradation suggests that an excessive focus on current task information destabilizes learning, leading to overfitting to new data and rapid forgetting of previously acquired knowledge. The optimal performance is typically achieved with balanced hyperparameter settings (e.g., $\eta = 2$, $\beta = 0.8$),

| Dataset | $\eta$ | $\beta$ | $\mathcal{A}_{\text{AUC}}$ | $\mathcal{A}_{\text{Avg}}$ | $\mathcal{A}_{\text{Last}}$ |
|---------|--------|---------|----------|----------|-----------|
|  | 2 | 0.8 | **68.16** | **67.36** | **71.32** |
|  | 20 | 0.8 | 67.66 | 67.22 | 71.01 |
|  | 100 | 0.8 | 66.45 | 67.05 | 70.96 |
| **CIFAR-100** | 2 | 1 | 67.98 | 67.34 | 70.99 |
|  | 2 | 20 | 66.58 | 66.94 | 70.80 |
|  | 2 | 100 | 66.13 | 66.75 | 70.17 |
|  | 2 | 0.8 | **46.07** | **51.08** | **43.43** |
|  | 20 | 0.8 | 45.99 | 51.01 | 43.20 |
|  | 100 | 0.8 | 45.55 | 50.92 | 43.14 |
| **ImageNet-R** | 2 | 1 | 45.95 | 50.95 | 43.20 |
|  | 2 | 20 | 44.34 | 49.12 | 42.95 |
|  | 2 | 100 | 43.87 | 48.36 | 41.90 |
|  | 2 | 0.8 | **66.79** | **72.01** | **64.08** |
|  | 20 | 0.8 | 66.28 | 71.38 | 64.10 |
|  | 100 | 0.8 | 66.32 | 71.40 | 64.14 |
| **Tiny-ImageNet** | 2 | 1 | 65.82 | 71.23 | 64.11 |
|  | 2 | 20 | 65.60 | 71.16 | 64.09 |
|  | 2 | 100 | 65.53 | 71.16 | 64.08 |

Table 10: AACL performance across varying hyperparameter settings: $\eta$ (Fisher weighting) and $\beta$ (current task importance) on multiple datasets, evaluated using $\mathcal{A}_{\text{AUC}}$, $\mathcal{A}_{\text{Avg}}$, and $\mathcal{A}_{\text{Last}}$ metrics.

| $\gamma$ | CIFAR-100 (%) | | | ImageNet-R (%) | | | Tiny-ImageNet (%) | | |
|----------|-------------|----------|-----------|--------------|----------|-----------|-----------------|----------|-----------|
|  | $\mathcal{A}_{\text{AUC}}$ | $\mathcal{A}_{\text{Avg}}$ | $\mathcal{A}_{\text{Last}}$ | $\mathcal{A}_{\text{AUC}}$ | $\mathcal{A}_{\text{Avg}}$ | $\mathcal{A}_{\text{Last}}$ | $\mathcal{A}_{\text{AUC}}$ | $\mathcal{A}_{\text{Avg}}$ | $\mathcal{A}_{\text{Last}}$ |
| 10 | 65.47 | 66.96 | 71.03 | 43.55 | 48.51 | 41.57 | 64.56 | 71.06 | 63.97 |
| 20 | 65.56 | 67.40 | 72.01 | 45.44 | 49.31 | 42.77 | 65.04 | 71.03 | 63.01 |
| 50 | 67.09 | 67.40 | 71.08 | 45.55 | 50.95 | 42.97 | 65.14 | 71.23 | 64.11 |
| 100 | **68.16** | **67.36** | 71.32 | **46.07** | **51.08** | **43.43** | **66.94** | **71.43** | 64.11 |
| 200 | 67.77 | 67.21 | **71.40** | 45.52 | 50.83 | 42.97 | 65.84 | 71.23 | **64.14** |
| 500 | 66.67 | 67.11 | 71.15 | 45.02 | 49.83 | 42.97 | 66.13 | 71.19 | 64.01 |
| 1000 | 66.06 | 66.76 | 70.51 | 44.87 | 48.99 | 41.92 | 65.97 | 70.45 | 63.04 |

Table 11: Effect of the regularization strength $\gamma$ on model performance across CIFAR-100, ImageNet-R, and Tiny-ImageNet under the Si-Blurry setting. Results are reported as percentages with seed 5.

which validates the critical importance of carefully tuning these parameters to maintain an effective trade-off between stability (retaining past knowledge) and plasticity (adapting to new information).

**Impact of $\gamma$ on covariance matrix** $R^{(0)}$**.** The parameter $\gamma$ in our algorithm initializes the posterior covariance matrix as $\mathbf{R}^0 = \gamma\mathbf{I}$, thereby directly controlling the model's regularization strength at the start of continual learning. As shown in Table 11, the choice of $\gamma$ influences performance across all three datasets under the Si-Blurry setting. High values (e.g., $\gamma = 1000$) impose excessive regularization, leading to underfitting by limiting the model's ability to adapt and degrading its overall performance. On the other hand, moderate values, such as $\gamma = 100$, yield optimal results, consistently achieving the highest performance across key metrics, including $\mathcal{A}_{\text{AUC}}$ and $\mathcal{A}_{\text{Last}}$. This suggests that reduced regularization enhances the model's adaptability by allowing more flexible posterior updates using the data-dependent term $\mathbf{Z}_t^\top\mathbf{Y}_t$ in the posterior mean. However, small values (e.g., $\gamma = 20$) offer only a marginal difference, potentially introducing numerical instability. Overall, this ablation highlights the sensitivity of our method to the initialization of $\mathbf{R}$ via $\gamma$, emphasizing the need to carefully tune this parameter to balance stability and plasticity.

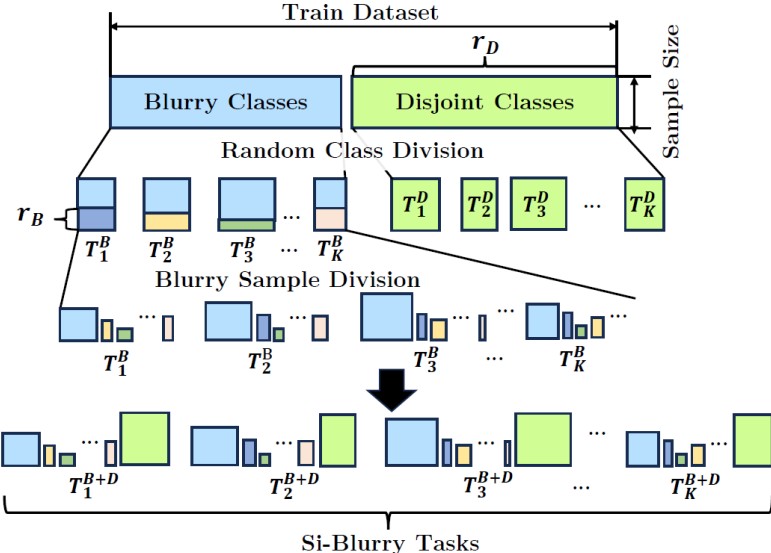

Figure 5: Overview of the Si-Blurry protocol Zhuang et al. (2024a), illustrating the process of dividing classes into disjoint and overlapping sets, followed by task assignment and sample redistribution across K tasks.

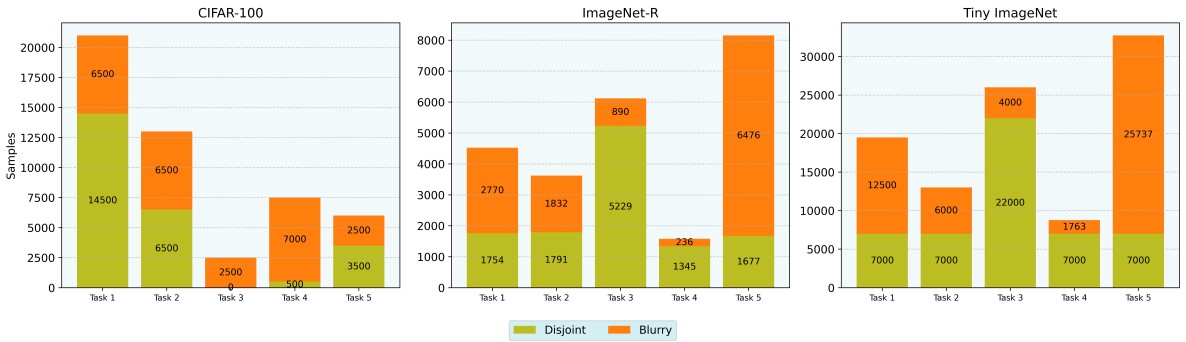

Figure 6: Sample distribution across five incrementally trained tasks for CIFAR-100, ImageNet-R, and Tiny-ImageNet datasets, highlighting the proportion of disjoint and blurry samples per task.

## A.4    Si-Blurry Setting

The Si-Blurry setting extends the standard blurry continual learning protocol by incorporating both shared and disjoint class distributions across tasks. Figure 6 depicts the sample distribution of the Blurry and disjoint classes for each task. This framework adheres to the fundamental properties of GCIL Mi et al. (2020), making it a robust benchmark for practical continual learning evaluation. As illustrated in Figure 5, this K-task learning approach operates through a systematic multi-stage process. Initially, the framework randomly divides all available classes into two distinct categories: disjoint classes, which remain unique to individual tasks, and overlapping classes, which may appear across multiple tasks. This division is governed by the disjoint class ratio $r_D$, representing the proportion of exclusive classes relative to the total class count. Following this classification, exclusive classes are allocated to dedicated tasks ($T^D$), while overlapping classes are distributed among shared tasks ($T^B$). Subsequently, each shared task undergoes a sample redistribution process, in which a portion of its samples is randomly transferred to other shared tasks. This redistribution is controlled by the blurry sample ratio $r^B$, which quantifies the fraction of redistributed samples within the complete set of shared task samples. The final Si-Blurry configuration, $T^{B+D}$, emerges as a hybrid structure combining both exclusive and shared task components, characterized by flexible task boundaries. To ensure

comprehensive empirical validation, we implement Si-Blurry using various combinations of the $r^D$ and $r^B$ parameters, enabling thorough performance assessment across different scenarios.

