# OpenReview forum: "Plasticity by Precision: Exemplar-free Analytic Adaptation for Class-Incremental Learning"
_TMLR — Under review for TMLR_

### Review · Reviewer_DzdX · 2026-05-29

**Summary Of Contributions:**

This paper introduces a new Analytic Continual Learning (ACL) framework, based on Elastic Weight Consolidation (EWC) and Analytic Class-Incremental Learning (ACIL). They do so, by merging ideas of these 2 different methods, namely, using a random projection layer over features of a frozen backbone (ACIL) and adding a Fisher Information Matrix term to the linear classification weights (EWC). The authors demonstrate the superiority of their method over a wide range of methods in the CIFAR-100, ImageNet-R and Tiny-ImageNet datasets.

**Audience:**

Yes

**Audience Explanation:**

Continual Learning is an important and active topic of research within the ML community, so it fits well within the scope of TMLR.

**Claims And Evidence:**

Yes

**Claims Explanation:**

The experiments done by the authors are comprehensive, and the datasets are relevant in vision.

**Requested Changes:**

I have a few criticisms with respect the paper presentation,

(1) The authors take inspiration from Bayesian regression. Even in the introduction, the authors write,

> The scope of this research focuses on class-incremental continual learning, where AACL integrates Fisher based regularization with Bayesian linear regression to derive a closed-form posterior update rule. Unlike conventional Bayesian models that rely on gradient-based inference or variational approximations, AACL employs analytic precision-weighted updates that follow the weight-invariant property of ACL (Zhuang et al. (2024a); Momeni et al. (2025)), where the recursively accumulated precision matrix and prior capture statistics bound to the backbone’s embedding distribution.

which seems to hint at a Bayesian formulation of their method. Furthermore, in the context of equation (7), the authors seem to hint at Bayesian statistics concepts, such as *prior knowledge*. Nonetheless, no Bayesian treatment is provided. Especially, the probabilistic treatment presented in section 3.1 is insufficient. In my view, the authors should either drop the Bayesian framing, or adopt a proper probabilistic treatment. If they choose the latter, here is a list of requested changes,

**Change 1.** The authors should make an effort to make the notions of "prior", "posterior", and "precision" meaningful. As far as I could parse through the authors treatment, $W$ is a random variable and seems to be drawn from $\mathcal{N}(0, \gamma I)$. The posterior is $p(Y|Z,W)$ which seems to follow a Gaussian model as well, with $p(Y|Z,W) = \mathcal{N}(ZW, \beta^{-1}I)$. $R^{t}$ and $(P^{t})^{-1}$ seem to be the posterior covariance of $W$ and its posterior precision, respectively. Note that none of this is formally stated, and the reader must infer it. An useful reference seems to be [R1], which the authors cite, who give a proper Bayesian treatment and whose Appendix F seems to be related to the proposed method

**Change 2.** Equation 6, in specific, is very troubling and should be clarified.

(2.a) Whenever taking expectations, the variable that it is taken with respect to must be explicit.

(2.b) assuming $W^{(t-1)} \in \mathbb{R}^{D \times C}$ (a matrix-valued random variable), $(W^{(t-1)})^{2}$ would be most naturally read as entry-wise power 2. The expectation $E[(W^{(t-1)}^{2})]$ is then another matrix. Therefore I have a hard time seeing how the 2nd equation relates to the 3rd in the chain in eq. 6.

(2.c) I have some trouble connecting $\lambda_{\text{Fisher}}$ (the term "Fisher" should be capitalized, BTW) to the actual Fisher Information Matrix (FIM) definition,

$$\mathbb{E}_{y\ \sim p(y|x,\theta)}[\nabla\_{\theta}\log p\_{\theta}\nabla\_{\theta}\log p\_{\theta}^{\top}]$$

could the authors clarify and desirably show that their eq. 6 is the actual diagonal of the FIM? That would require expliciting the posterior distribution and links back to my requested change 1.

# References

[R1] Nguyen, Cuong V., et al. "Variational continual learning." arXiv preprint arXiv:1710.10628 (2017).

---

### Review · Reviewer_zDWd · 2026-07-03

**Summary Of Contributions:**

This paper proposes a continual learning algorithm for the class-incremental setting. The core of the method is a set of precision and classifier weight matrices that are maintained and updated across tasks through a Bayesian-inspired analytic update procedure. Without maintaining replay buffers or performing iterative retraining, the proposed method achieves excellent performance across a diverse set of benchmark tasks.

**Audience:**

Yes

**Audience Explanation:**

Continual learning is an important research area, and the audience will welcome a method that is simple and effective.

**Broader Impact Concerns:**

I don't see any broader impact concern from the current submission.

**Claims And Evidence:**

No

**Claims Explanation:**

The main algorithmic component relies on several heuristic design choices and approximations that lack rigorous theoretical justification. For instance, Equation (6) approximates the Fisher information using the squared magnitudes of the classifier weights, while Equations (7)–(9) introduce update rules for the precision and weight matrices. In particular, the updates in Equations (7)–(9) resemble those of Bayesian linear regression, but it is unclear from which probabilistic model they are derived, or whether they correspond to a principled Bayesian posterior update. The experimental evaluation also does not directly justify these design choices. While the paper reports strong final performance and includes several ablation studies, it does not provide sufficient evidence or analysis explaining why these particular approximations and update rules are appropriate.

**Requested Changes:**

- Please provide a clearer theoretical justification for the heuristic design choices underlying the main algorithm, especially Equations (7)–(9). In their current form, these update rules appear largely heuristic. While they resemble the posterior updates in Bayesian linear regression, it is unclear from which probabilistic model they are derived or what approximations are required to obtain the proposed update equations.
- The paper emphasizes the advantage of being "exemplar-free." However, it would be valuable to discuss the actual memory cost of the proposed implementation and compare it fairly with replay-based methods. Although the method does not store historical exemplars, it maintains high-dimensional statistical state (e.g., precision/covariance matrices and classifier weights), whose memory cost may be substantial. In particular, the D×D precision/covariance matrices can become considerably larger than the memory required by replay methods storing a moderate number of latent features or exemplars. A comparison in terms of actual memory consumption (e.g., bytes) would make the claimed advantage more convincing.

---

### Review · Reviewer_zWL6 · 2026-07-17

**Summary Of Contributions:**

- The paper proposes AACL, an exemplar-free analytic method for class-incremental learning.
- The method uses a frozen backbone and updates the classifier in closed form.
- The precision matrix combines prior information, a weight-based importance term, and statistics from the current data.
- The paper evaluates the method on CIFAR-100, ImageNet-R, Tiny-ImageNet, and ImageNet-1000.
- The main strengths are the exemplar-free setup, low training cost, several ablation studies, and experiments with different backbones.
- The main weakness is that the novelty over closely related analytic methods is not explained clearly enough.
- There are also several inconsistencies in the results, claims, tables, and presentation.

**Additional Comments:**

- The main idea is interesting, but the current version is difficult to evaluate because the novelty is not positioned clearly against the closest prior work.
- The paper should make the technical difference to existing analytic continual learning methods much more explicit.
- At the moment, it is hard to judge whether this is a major methodological contribution or a smaller extension of an existing analytic update.
- The manuscript also needs stronger quality control. Several numerical, textual, bibliographic, and visual inconsistencies reduce confidence in the results.

**Audience:**

Yes

**Audience Explanation:**

- Exemplar-free continual learning is relevant to the TMLR audience.
- Analytic updates without replay or backpropagation can be useful when memory, privacy, or computation is limited.
- The results suggest that the method may be competitive in blurry class-incremental settings.
- The efficiency experiments and the use of several backbones are also potentially useful.
- However, the paper needs a clearer novelty argument and a more consistent evaluation before the contribution can be assessed reliably.

**Broader Impact Concerns:**

- I do not see a major broader impact concern that requires a separate statement.
- The paper discusses privacy benefits from not storing previous examples, which is reasonable.
- However, the privacy claims should remain limited. Exemplar-free learning does not by itself guarantee privacy.
- The claims about practical deployment should also be stated carefully because the experiments are limited to standard image benchmarks.

**Claims And Evidence:**

No

**Claims Explanation:**

- Some important claims are not consistent with the reported results.
- The abstract and conclusion report gains of 8 percent, 3 percent, and 4 percent, while the result section gives 8 percent, 4 percent, and 3 percent.
- The paper claims consistent superiority over all baselines, but AACL is worse than MVP-R for some ImageNet-R metrics.
- Table 2 contains a suspicious ImageNet-R value of 60.01 for buffer size 7000. The surrounding values are close to 50, so this may be a typo.
- Table 1 reports mean and standard deviation over five runs, while several other tables report only the best result from five runs. This makes the evaluation inconsistent.
- The paper calls squared classifier weights a Fisher information approximation, but this connection is not derived or justified clearly.
- The Bayesian interpretation is also not developed in enough detail. The prior, likelihood, and assumptions should be stated clearly.
- The main text presents the Woodbury update as almost equivalent to the original update, but the appendix introduces an additional approximation.
- The task-agnostic claim is unclear because the algorithm and experiments still use explicit task sequences.
- The architecture-agnostic claim is too strong because the paper only tests AACL with several backbones. It does not compare competing methods using the same backbones.
- Because of these issues, the main conclusions are not yet fully supported.

**Requested Changes:**

Critical changes
- Correct all inconsistent result numbers in the abstract, conclusion, tables, and result discussion.
- Check the suspicious value in Table 2 and verify all table entries.
- Remove or revise claims of consistent superiority when the reported results do not support them.
- Report results consistently. Please use mean and standard deviation over the same seeds for all main comparisons instead of mixing averages with best-of-five results.
- Explain the novelty much more clearly in the Related Work section, ie discuss, please!
- The paper should state which parts are taken from ACIL, RanPAC, GACL, Bayesian linear regression, or recursive least squares.
- The paper should explain exactly what is new in AACL.
- It is currently unclear whether the main novelty is the complete update, the squared-weight regularizer, or only the use of the method in the Si-Blurry setting.
- Add a direct technical comparison with the closest methods, especially ACIL, RanPAC, GACL, AnaCP, and MoAL.
- A comparison table would be useful. It could include exemplar use, gradient use, task boundary requirements, overlapping classes, covariance type, feature adaptation, and update rule.
- Add a standard recursive least squares or Bayesian linear regression baseline.
- Add a component-wise ablation of the precision update.
- The ablation should separately test prior precision, current-data covariance, and the squared-weight importance term.
- Justify why squared weights can be interpreted as Fisher information. Otherwise, use a more careful name such as weight-based importance.
- Provide a clearer probabilistic derivation of the claimed Gaussian posterior and Bayesian update.
- Clarify the Woodbury section. Separate the exact Woodbury identity from the additional approximation used for the inverse.
- Provide a complete complexity analysis that includes feature dimension, batch size, rank, and number of classes.
- Clarify whether task boundaries are known during training.
- If the method is claimed to be task-agnostic or boundary-free, evaluate it in a truly boundary-free stream. Otherwise, reduce this claim.
- Add matched baseline results for the different backbones, or reduce the architecture-agnostic claim.
- Consider adding an offline joint-training reference as an upper bound, especially because the paper claims exact preservation of previous knowledge.

Changes that would strengthen the paper
- Improve the Related Work discussion. It currently lists methods but does not explain the technical differences well enough.
- Explain why GACL does not already solve the main problem addressed by AACL.
- Include training time, inference time, and memory comparisons with analytic baselines such as ACIL, RanPAC, and GACL.
- Check every bibliography entry against the final publisher version.
- Add DOI information where a DOI is available, ie almost everywhere.
- Use final published versions instead of preprints where possible.
- Use consistent fonts in all figures.
- Figures 2, 3, and 4 use typography that is different from the main text.
- Increase the text size inside the figures.
- Regenerate Figure 5 with larger labels and preferably as vector graphics.
- Improve Table 1 layout. The Mem Size header and group values could be rotated or placed vertically to create more space for the numerical columns.
- Use consistent font sizes across all tables.
- Table 10 uses a noticeably larger font than most other tables.
- Improve spacing, decimal alignment, and separation between dataset blocks.
- Perform a full language and formatting check. There are several small grammar, spacing, and consistency issues.